# DreamLight: Towards Harmonious and Consistent Image Relighting

**Yong Liu**[1*]**, Wenpeng Xiao**[2*]**, Qianqian Wang**[2] **, Junlin Chen**[2] **, Shiyin Wang**[2] **,**
**Yitong Wang**[2] **, Xinglong Wu**[2] **, Yansong Tang**[1†]

[1]Tsinghua Shenzhen International Graduate School, Tsinghua University  [2]ByteDance Inc.

`liuyong23@mails.tsinghua.edu.cn, tang.yansong@sz.tsinghua.edu.cn`

## Abstract

We introduce a model named DreamLight for universal image relighting in this work, which can seamlessly composite subjects into a new background while maintaining aesthetic uniformity in terms of lighting and color tone. The background can be specified by natural images (image-based relighting) or generated from unlimited text prompts (text-based relighting). Existing studies primarily focus on image-based relighting, while with scant exploration into text-based scenarios. Some works employ intricate disentanglement pipeline designs relying on environment maps to provide relevant information, which grapples with the expensive data cost required for intrinsic decomposition and light source. Other methods take this task as an image translation problem and perform pixel-level transformation with autoencoder architecture. While these methods have achieved decent harmonization effects, they struggle to generate realistic and natural light interaction effects between the foreground and background. To alleviate these challenges, we reorganize the input data into a unified format and leverage the semantic prior provided by the pretrained diffusion model to facilitate the generation of natural results. Moreover, we propose a Position-Guided Light Adapter (PGLA) that condenses light information from different directions in the background into designed light query embeddings, and modulates the foreground with direction-biased masked attention. In addition, we present a post-processing module named Spectral Foreground Fixer (SFF) to adaptively reorganize different frequency components of subject and relighted background, which helps enhance the consistency of foreground appearance. Extensive comparisons and user study demonstrate that our DreamLight achieves remarkable relighting performance.

## 1 Introduction

With the rapid development of generative models in recent years [1, 2, 3, 4], image composition has received increasing attention owing to its capacity for controlled generation [5, 6, 7]. However, since the implanted foreground and the new background originate from different sources, this discrepancy can easily lead to an unrealistic perception of the composite image. In this paper, we focus on a challenging task named universal image relighting, which aims to seamlessly composite a subject into a new background while maintaining realism and aesthetic uniformity in terms of lighting and color tone. The background can either be specified by provided natural images (image-based relighting), or be created from unlimited text prompts (text-based relighting). This task ensures that users and digital elements coexist naturally within any environment and has wide application in virtual reality and intelligent editing for the film and advertising industries [8, 9, 10, 11, 12, 13].

---

[*]Equal Contribution
[†]Corresponding author

39th Conference on Neural Information Processing Systems (NeurIPS 2025).

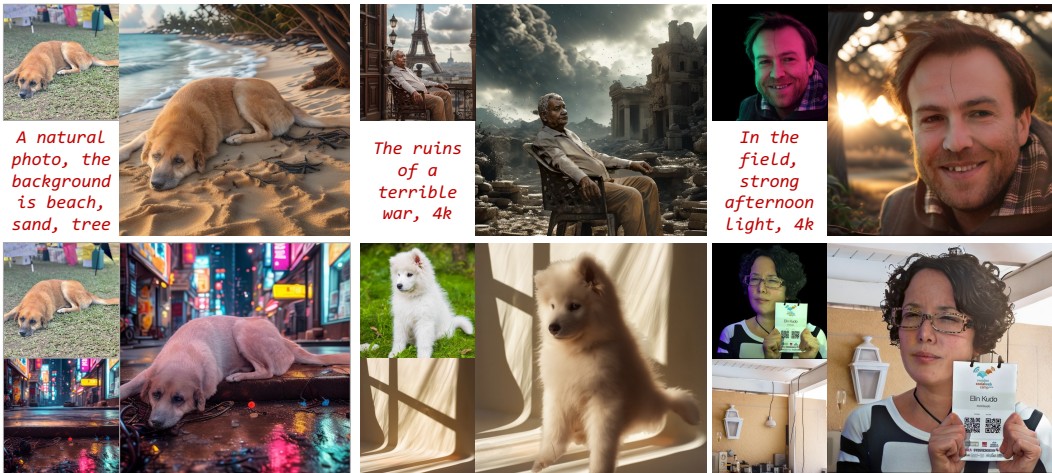

Figure 1: Our DreamLight is a unified image relighting model capable of performing relighting guided by arbitrary text or natural background images without any other prior such as HDR maps. The top left of each set of figures is the original picture of the foreground, the bottom left is the given condition, and the right is the generated result.

Early researches about relighting tend to focus on the image condition, *e.g.*, image harmonization and portrait relighting, while with scant exploration into text-based scenarios. Portrait relighting methods [14, 15, 16, 17] mainly take the idea of physics-guided design that explicitly models the image intrinsics and formation physics. They decompose the input images into several components and leverage a phased pipeline to incrementally learn image components such as surface normals and albedo maps. Some harmonization methods [18, 19] also take similar disentanglement idea. Despite the promising results achieved by these approaches, obtaining training data with pairs of high-quality relighting images and their corresponding intrinsic attributes from light stage is expensive and difficult. Most harmonization methods [20, 21, 22, 23] take this task as an image-to-image translation paradigm and propose to perform pixel-level transformation based on autoencoder architecture. However, they lack object semantic guidance and tend to overlook distinct variations in illumination. Recently, some works [24, 8, 25] exploit the powerful semantic modeling ability of generative diffusion models [1] to enhance relighting effect, but their light source relies on the given environment maps and such maps are not always feasible to acquire in real world. IC-Light [26] is proposed to address both image-based and text-based relighting for natural images without introducing other signal guidance. It imposes light alignment training strategy to enhance the light consistence. However, IC-Light takes two separate models for different conditions and does not tailor the structure. Simply concatenating foreground, background, and noise limits the model's understanding of foreground-background interactions, leading to severe color bleeding and foreground distortions in some scenarios.

To address the above challenges, we present a model named DreamLight that can perform both image-based and text-based relighting. In DreamLight, to generate natural interaction effects of light and color tone between the foreground and the background, we propose a Position-Guided Light Adapter (PGLA), which condenses light information from different directions in the background into several groups of light query embeddings and selectively modulates the foreground area with direction-biased masked attention. In addition, we present an effective Spectral Foreground Fixer (SFF) as a post-processing module to enhance the consistency of foreground appearance and avoid subject distortion. Based on the wavelet transform, SFF is trained to learn dynamic calibration coefficients for high-frequency textures of input foreground and relighted low-frequency light. By adaptively reorganize these information, SFF can output stunningly consistent foreground. Besides, to facilitate the training of our model, we develop different kinds of data generation processes, *e.g.*, 3D rendering and training relighting lora, to produce diverse training samples.

We have performed extensive quantitative and qualitative comparisons. Experiment results demonstrate that our DreamLight exhibits superior generalization and performance on the universal relighting of natural images. Related ablations also prove the rationality and effectiveness of the proposed designs. Furthermore, we observe that thanks to the unified learning of text and image conditions in a single model, our DreamLight can generate results with the guidance of both conditions.

Our contributions can be summarized as follows:

- We propose a model named DreamLight for universal image relighting, which can seamlessly composite a subject into a new background with either image or text conditions. We also develop high-quality data generation process to benefit the training of our model.

- We introduce a Position-Guided Light Adapter (PGLA), which is designed to enable foreground elements at different positions to interact with background light from various directions in a tendentious manner for generating more natural lighting effects.

- We propose a Spectral Foreground Fixer (SFF) module to adaptively reorganize different frequency components of the input and relighted subjects, which helps enhance the consistency of foreground textures.

## 2 Related Work

**Image-based Relighting:** There are two principal sets of related image-based relighting methods. The first is portrait relighting approaches [15, 14, 16, 27, 28, 29, 30, 8, 17, 31]. They mainly take the idea of physics-guided model design that typically involve the intermediate prediction of surface normals, albedo, and a set of diffuse and specular maps with ground truth supervision. To achieve that, some methods rely on the paired training data acquired with the light stage system [32] and a target HDR environment map as the external light source. However, the dependence on light stage data and HDR maps incurs substantial data collection cost and significantly limits their implementation in real-world situations, where obtaining HDR maps may not always be feasible. Some methods [33, 31] employ multi-stage frameworks. As a result, the accuracy and performance of these systems hinge on the precision of each individual stage. This makes the entire process complex and susceptible to errors that could propagate throughout these intermediate steps. The second is image harmonization methods that aim to match the color statistics of the foreground object with those of the background for natural composition [18, 20, 22, 23, 34, 21, 19, 35, 36, 37, 38]. These methods tend to take this task as an end-to-end image-to-image translation paradigm, where the network is trained to predict a harmonized image from the input composite. Some works [39, 40] collect pixel-aligned paired data by color transfer or altering foreground color in real images with pre-designed or learned augmentations. While these methods have achieved decent harmonization effects, they struggle to generate realistic and natural light interaction effects between the foreground and the background. Recently, some works [24, 8, 25] exploit generative models [1] to enhance relighting effect, but most of them still rely on the given environment maps for lighting guidance, which limits the potential application scenarios. IC-Light [26] proposes to perform pure natural image relighting by imposing light alignment training strategy and achieves excellent performance for scenarios with strong lighting. However, its simple network design limits the model's performance and is prone to generating severe color bleeding. To alleviate above issues, we propose DreamLight designed for natural images without any additional prior source. A position-guided light adapter is introduced to boost the reasonable interaction between subjects and backgrounds, thus contributing to generating natural and harmonious lighting effect.

**Text-based Relighting:** Currently, there is relatively less research focused on text-based relighting. Although Lasagna [41] takes text as input, the text serves to indicate the light direction rather than to describe the target background. The pioneer IC-Light [26] utilizes two separate models to handle image-based and text-based relighting. Actually, some text-guided inpainting methods [42, 43] can achieve similar effect. However, they tend to preserve the original lighting and color of the subject, which may result in unnatural results. Conversely, our DreamLight achieves powerful relighting performance of natural images using a single model for both image-based and text-based situations.

## 3 Method

### 3.1 Overview

Figure 2 illustrates the pipeline of our DreamLight. It takes a triplet of foreground image, background image, and text prompt as input. For image-based relighting, the text prompt is set to "blend these two images". For text-based relighting, the background image is designed as an all-black image. The **green** and **brown** lines show the specific processes of image-based and text-based relighting, respectively. In detail, we first utilize a pretrained segmentation model [26, 44] to extract the subject

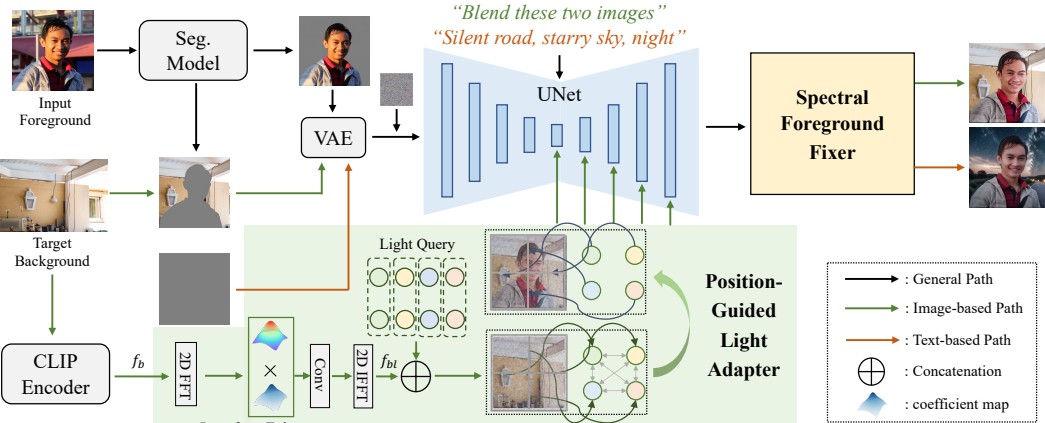

Figure 2: Pipeline of our DreamLight. It takes the foreground image, background image, and text prompt as input. The masked foreground and background images are encoded by pretrained VAE and concatenated with the random noise to serve as the input of UNet. The Position-Guided Light Adapter is proposed to selectively inject background light from different directions into the foreground at different locations for more natural relighting results. The Spectral Foreground Fixer is utilized to enhance the consistency of the subject.

region and remove the original background from the input image. Following [45, 26], the masked subject and background are encoded by VAE and then concatenated with the random noise to serve as the input for diffusion UNet. For image-based relighting, a Position-Guided Light Adapter (PGLA) is proposed to selectively inject background light information from different directions into the foreground with direction-biased masked attention. Finally, a Spectral Foreground Fixer (SFF) is utilized to enhance the consistency of the foreground.

## 3.2  Position-Guided Light Adapter

Although simply concatenating the background with random noise can convey some information about environment lighting, it imposes pixel-aligned light prior and overlooks the natural interaction between the subject and light from various directions in the background, thus resulting in undesirable relighting results in some scenes. To alleviate this problem, we propose the Position-Guided Light Adapter (PGLA), which enhances the foreground's response to light sources from different directions in background while reducing potential unreasonable light alignment. This is achieved by additional encoding and organization of the background light information.

Specifically, the overall pipeline of our PGLA is based on the process of IP-Adapter [46]. We first utilize a CLIP image encoder to encode the target background image into a feature map $f_b \in \mathbb{R}^{H \times W \times C}$, where $C$ indicate the embedding dimension. $H, W$ denote the spatial size of feature map. To mitigate the potential noise effects of background textures on light information extraction, we perform a low-frequency enhancement operation on the encoded background features, which helps emphasize the high-level lighting and color tones information. The process can be formulated as:

$$
\begin{aligned}
g &= Gaussian(H, W, \sigma), \\
f_{bl} &= FFT(f_b) * g, \\
f_{bl} &= IFFT(ReLU(Conv(f_{bl}))) + f_b,
\end{aligned}
\tag{1}
$$

where $FFT$ and $IFFT$ are Fourier transform and Fourier inverse transform. $g$ denotes the filtering coefficient map. $\sigma$ is cutoff frequency and $*$ means element-wise product. $Conv$ and $ReLU$ operations are utilized to update the features in the spectral domain for efficient global interaction.

Then, we restructure the background features and encode light information from different directions into predefined light queries $f_Q$, which are randomly initialized learnable embeddings. Considering that the condition is vanilla 2D natural background images, we assume that the light sources can be split into four basic directions: left, right, top, and down. Thus, we leverage four sets of query embeddings to selectively extract light information from background features.

To achieve that, we propose a direction-biased masked attention. As shown in Figure 3, we use cross attention mechanism to condense the information of background into the light query. In detail, we initialize four sets of light query and design them to separately perceive the background light sources of the four directions, i.e., left, right, top, and down. Taking the query $f_Q^{left}$ corresponding to the left light as an example, we generate a coefficient map that decays from left to right with the same size as the background feature, as shown by the "left decay map" in Figure 3. It is then flattened and multiplied with the initial attention weight of corresponding regions to modulate the cross attention, making the query $f_Q^{left}$ focus more on the information on the left side of the background feature. Similarly for the light query responsible for the other directions. Besides, we concatenate the light query and background features as Key and Value to ensure the interaction between different group of queries, which contributes to the overall harmonization.

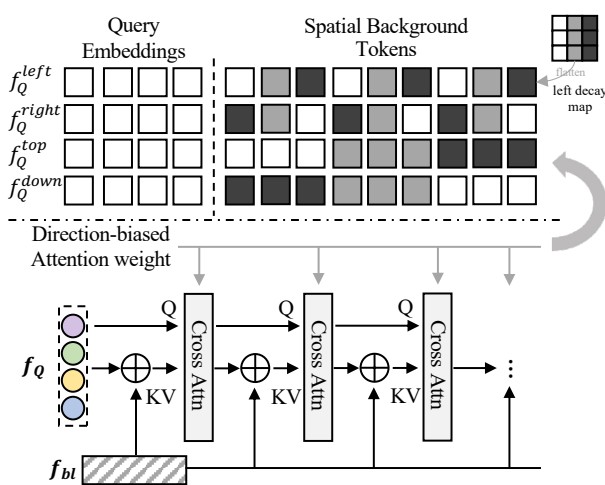

Figure 3: Process of direction-biased masked attention. The upper figure is the design of the attention mask in cross attention. Here the darker color indicates the tendency toward 0. To allow different groups of query to perceive different directions of light, we generate coefficient maps with different attenuation directions and multiply them to the original attention weight. For ease of understanding, we note the transformation of the left decay map in the figure. Besides, the attention weight between different queries has not been modified for overall light information harmonization.

Having these condensed light queries, the next step is to inject their information into the foreground area of the latent features in UNet with similar masked attention. That is, we adjust the attention weight of condition cross attention so that different regions of the foreground object have a stronger response to nearby light sources. In addition to the exchange of Q and KV, we additionally add a mask to the background area to ensure that the background is not changed. Furthermore, as shown in Figure 2, we only inject these light prior in the middle and up block of the UNet. This is due to the fact that feature changes in down blocks may have an impact on the overall semantics [47]. We only need to make changes to the object's light based on its overall representation, which helps to avoid the potential distortion problems caused by extra information injection.

With such designs, the combined subject elements can perform selective interaction with the background to produce more natural relighting results. Related experiments in Section 4.3 also justify the rationality of our designs.

## 3.3 Spectral Foreground Fixer

Diffusion-based methods tend to face the problem of foreground distortion, especially in small and detailed areas such as face and text. On the one hand, the latent space of large-scale pre-trained models tends to embellish the foreground, which can lead to inconsistency and ID variation with the input. On the other hand, the encoding process of VAE may cause information loss in small regions with high information density, leading to difficulties in maintaining the texture of the subject. Therefore, we propose a Spectral Foreground Fixer (SFF) to address this challenge. This module is based on the assumption that the high-frequency components of an image correspond to the pixels varying drastically, such as object boundaries and textures, while the low-frequency components correspond to the general semantic information such as the color and light.

As shown in Figure 4, we utilize Wavelet Transform to extract the high-frequency and low-frequency components of the input foreground image and the initial predicted results. It can be seen that the high-frequency part maintains the details and textures, while low-frequency part indicate rough colors and tones. Then we combine the high-frequency part of input foreground image with the low-frequency component of the initial relighting result and feed them into a Modulator, which takes

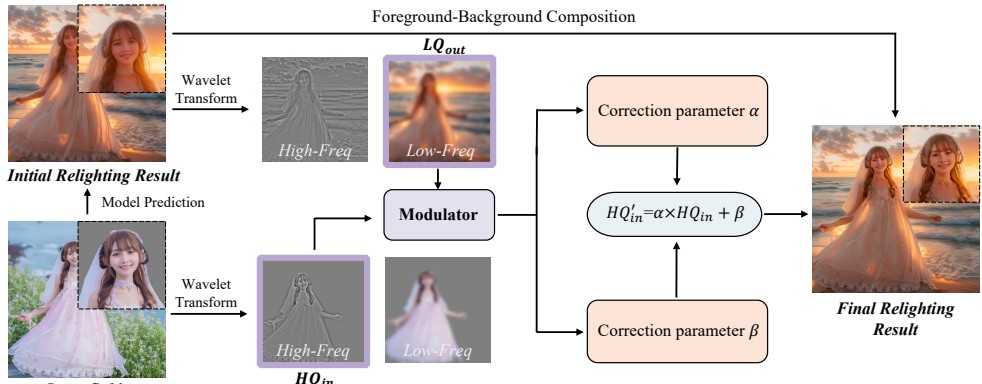

Figure 4: Process of Spectral Foreground Fixer. The modulator is utilized to predict a set of coefficients for modulating the textures of foreground with the light of background.

image-to-image generation paradigm and is trained to predict a set of coefficients to reorganize these information. The reorganization process can be formulated as:

$$
\begin{aligned}
\alpha, \beta &= \mathcal{M}(HQ_{in}, LQ_{out}), \\
HQ'_{in} &= HQ_{in} * \alpha + \beta, \\
I'_{out} &= HQ'_{in} + LQ_{out},
\end{aligned}
\tag{2}
$$

where $HQ$ and $LQ$ denote the high-frequency and low-frequency part extracted by Wavelet Transform, respectively. $\mathcal{M}$ is the modulator. $\alpha \in \mathbb{R}^{H \times W \times 3}$ and $\beta \in \mathbb{R}^{H \times W \times 3}$ are the predicted modulation coefficients. $\alpha$ is utilized to control the influence degree of foreground texture and $\beta$ contains the balance information for harmonizing the foreground. Compared to directly reorganize the foreground texture and background semantics, such design helps to output more consistent and natural results, avoiding artifacts from forced combination. Finally, we replace the foreground region of initial relighting images by the predicted $I'_{out}$ according to the foreground mask.

The modulator $M$ is individually trained in a self-supervised manner. Specifically, we perform random color transformation on arbitrary natural images to obtain pseudo pairs of relighting data, where the transformed image is input and original image serves as target. Then we extract the high-frequency component of the transformed image and the low-frequency component of the original image as inputs to the modulator for modulation coefficient prediction. The modulator is expected to adaptively combine the high-frequency and low-frequency parts from different source and avoid potential color noise or artifacts. We utilize MSE and perceptual loss [48] to supervise the learning of the modulator. Furthermore, to promote training stability and improve coordination, the supervision is applied on both the predicted high-frequency part $HQ'_{in}$ and entire output image $I'_{out}$.

## 3.4 Data Generation

We design data generation pipeline to facilitate the training of our model. Our data has three sources. Firstly, we construct pairs by training a relighting ominicontrol [49] lora in a bootstrapping manner, *i.e.*, training – incorporating results into training set – continue training. The initial set consists of 100 image pairs collected from time-lapse photography videos and self-photographed photos. After each training, the model is used to relight vanilla images. High quality pairs are selected and incorporated into the training set for continue training. We will open source this relighting lora to benefit community. Secondly, we utilize available 3D assets [50] to render a number of consistent images with lighting of different color and directions. We construct an automatically rendering pipeline on 3D Arnold Renderer, and generate various lighting effects with random light sources and HDR images for corresponding pairs. Finally, to enhance data diversity, we also process vanilla images with IC-Light [26] and filter out high-quality synthetic data pairs with aesthetic score [51]. The prompts are generated through a two-step process: GPT-4 [52] initially brainstorms over 200 fundamental scenarios, which are then tailored by LLaVA [53] according to the main subjects present in the images. Totally, the quantities of the three types of data are about 600k, 150k, and 300k, respectively. Please see the supplementary material for detailed analysis about the training data.

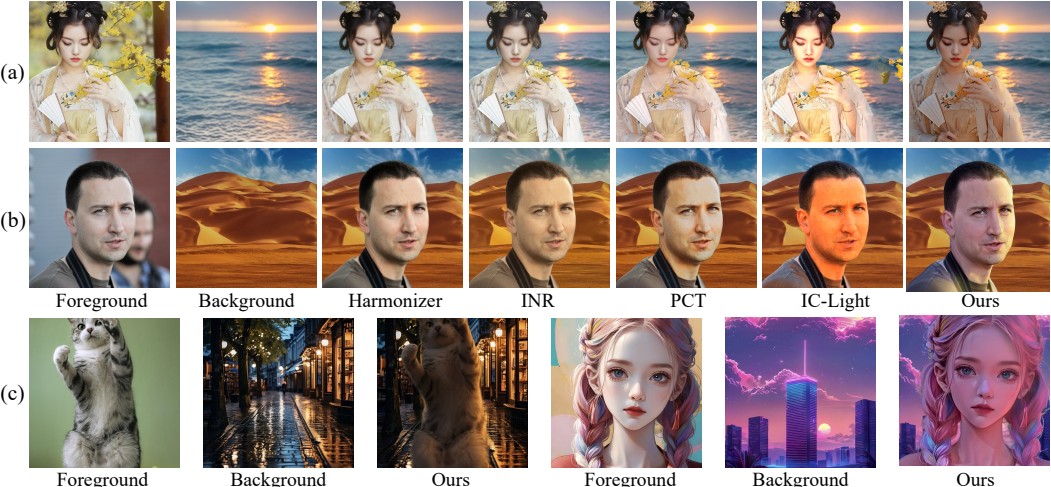

| Foreground | Background | Harmonizer | INR | PCT | IC-Light | Ours |

| Foreground | Background | Ours | Foreground | Background | Ours |

Figure 5: Image-based relighting results. (a) and (b) demonstrate the comparisons with popular available image-based relighting methods, *i.e.*, Harmonizer [38], INR [21], PCT [20], and IC-Light [26]. (c) shows the generalizability of our DreamLight to foregrounds of different categories and various styles of images. Please see the supplementary material for more qualitative results.

## 4 Experiment

### 4.1 Implementation Details

Our model is implemented in PyTorch [54] using $8 \times$ A100 at $512 \times 512$ resolutions. The main model and fixer model are trained separately. The main model is trained end-to-end with the batch size of 512. The learning rate is set to 5e-5. We leverage StableDiffusion-v1.5 [1] as the base generative model and CLIP-H [55] as the encoder for position-guided light adapter. Following [26], we takes RMBG-1.4 as the segmentation model for extracting the region of subject. The spectral foreground fixer is finetuned on the VAE model of StableDiffusion-v1.5. The cutoff frequency of spectral filter is set to 5 in default. The number of light query is set to 4 for each direction. The evaluation benchmark contains 600 high-quality image pairs rendered by Arnold Renderer from real objects. For image-based relighting, we take the popular metrics of standard Peak Signal-to-Noise Ratio (PSNR), Structural Similarity (SSIM), Learned Perceptual Image Patch Similarity (LPIPS) [56], and image similarity score calculated by CLIP [55] (CLIP-IS) to verify the effectiveness of our method. For text-based relighting, we utilize image-text matching CLIP score, aesthetic score [51], and Image Reward (IR) [57] score to assess the plausibility of the generated results. IR score is calculated by text-to-image human preference evaluation models trained on large-scale datasets of human preference choices. Aesthetic score is a linear model trained on image quality rating pairs of real images. Please refer to the supplementary material for more details and illustrations.

### 4.2 Main Results

**Qualitative Comparison:** In Figure 5 and Figure 6, we present relighting results of our method and the comparison with existing methods. Our DreamLight achieves excellent performance of both image-based and text-based relighting in a single model. More relighting results can be found in the supplementary materials. From (a) and (b) in Figure 5 we can see that our method not only harmonizes the lighting of the foreground and background, but also more effectively models the interaction between light sources and objects within images. In addition to human and natural backgrounds, we illustrate the generalization ability of our model for foregrounds of different categories and other stylistic images in (c). In Figure 6 we compare our method with IC-Light [26] and existing state-of-the-art prompt-guided inpainting methods since existing studies have paid limited attention to relighting based on given prompts. Although inpainting-based methods are capable of generating backgrounds that are in accordance with text prompts, they exhibit a propensity towards maintaining the color and lighting of the foreground unchanged, leading to an incongruity with the background. IC-Light, in contrast, grapples with issues of excessive color variations in

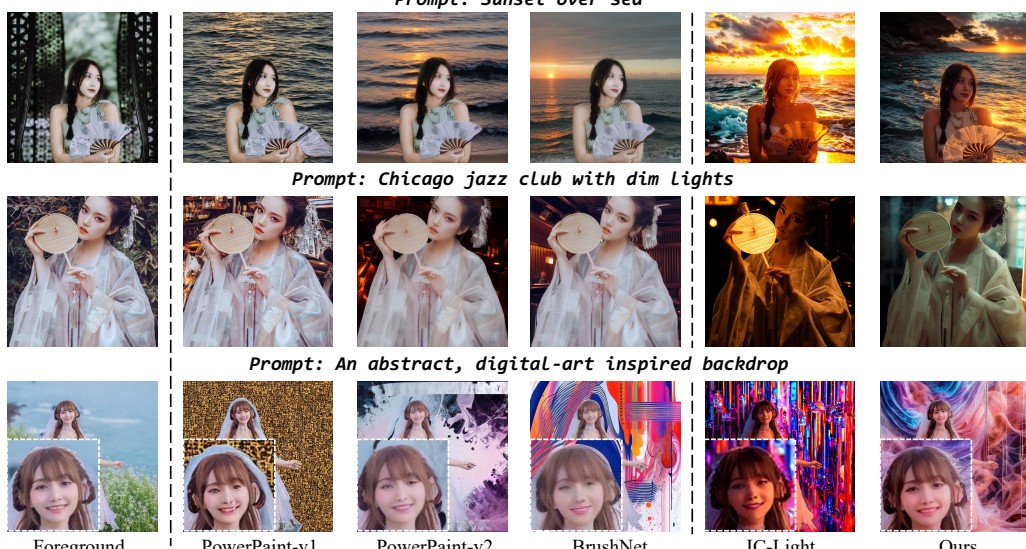

Figure 6: Results of text-based relighting. PowerPaint [43] and BrushNet [42] are powerful inpainting methods that can perform prompt-guided inpainting. To better illustrate, we demarcate the foreground, outpainting methods, and relighting methods with dashed lines.

| Method | PSNR↑ | SSIM↑ | LPIPS ↓ | CLIP-IS ↑ |
|---|---|---|---|---|
| INR [21] | 16.79 | 0.694 | 0.231 | 0.872 |
| PCT [20] | 18.45 | 0.714 | 0.206 | 0.895 |
| IH [19] | 20.66 | 0.762 | 0.177 | 0.896 |
| IC-Light [26] | 20.27 | 0.771 | 0.181 | 0.889 |
| Ours | **22.15** | **0.783** | **0.158** | **0.908** |

Table 1: Quantitative results comparison about image-based relighting.

| Method | CLIP Score↑ | Aesthetic Score↑ | IR Score↑ |
|---|---|---|---|
| PP-V1 [43] | 0.605 | 5.58 | 1.87 |
| PP-V2 [43] | 0.627 | 5.97 | 2.50 |
| BrushNet [42] | 0.613 | 5.73 | 1.96 |
| IC-Light [26] | 0.629 | 6.04 | 2.25 |
| Ours | **0.644** | **6.32** | **3.47** |

Table 2: Quantitative results about text-based relighting. PP denotes PowerPoint.

the generated images as well as distortions of the foreground. Our approach can produce relighting results with more natural light. Additionally, the bottom row illustrates the capacity of our method to maintain consistency for the subject. We can observe that all other methods generate results with significant facial distortion, whereas ours ensures excellent subject consistency.

**Quantitative Comparison:** Table 1 and Table 2 display the evaluation metrics of different methods about image-based and text-based relighting, respectively. Table 1 indicates that our approach can yield more consistent and harmonious outcomes when provided with a target background image. Results in Table 2 show that our DreamLight exhibits advantages in terms of vision-text compatibility, aesthetic appeal, and rationality. Results of user study are reported in the supplementary materials.

## 4.3 Case Study

**Position-Guided Light Adapter:** In Figure 7 we conduct visualization comparison of the ablations about the PGLA. As depicted in Figure 7, the model struggles to learn the light interaction between the foreground and background without any prior imposition (W/o adapter). When applying vanilla IP-Adapter [46], which performs arbitrary interaction between subject and background, information from diverse directions in the background interferes with each other, thereby affecting the final relighting

| Method | PSNR↑ | SSIM↑ | LPIPS↓ | CLIP-IS↑ |
|---|---|---|---|---|
| w/o adapter | 18.58 | 0.732 | 0.221 | 0.865 |
| vanilla IP.A. | 20.23 | 0.752 | 0.184 | 0.891 |
| w/o Filter | 21.81 | 0.778 | 0.162 | 0.903 |
| PGLA (Ours) | **22.15** | **0.783** | **0.158** | **0.908** |

Table 3: Results of different light adapter designs. "IP.A." means IP-Adapter [46]. Filter is the spectral filter used to enhance the low-frequency component of background.

performance. Through the utilization of direction-biased masked attention, the model selectively transmits background lighting information, enabling the foreground to acquire lighting that is harmonious with the background (the last two columns). The design of low-frequency enhancement further discards irrelevant information exist in the background textures, thus contributing to robust training

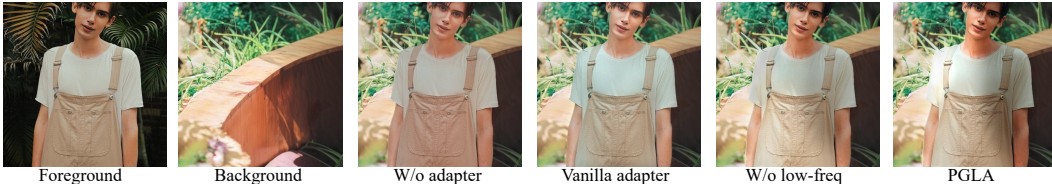

| Foreground | Background | W/o adapter | Vanilla adapter | W/o low-freq | PGLA |

Figure 7: Visualization comparisons with different light adapter design strategies.

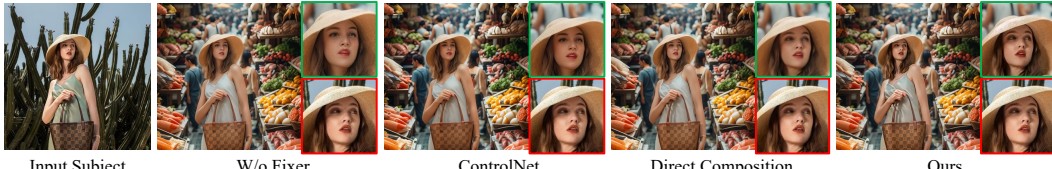

| Input Subject | W/o Fixer | ControlNet | Direct Composition | Ours |

Figure 8: Visualization comparisons with different foreground fixer strategies. For the ease of observation, we accentuate the changes in the facial region on the right side of each relighting result. *The green box above showcases the magnified facial region of the relighted results.* The red box below presents the magnified facial region of the input foreground image. Best viewed zoom-in.

of the model and promoting generation of more natural and consistent results. Quantitative results in Table 3 also demonstrate the rationality of our designs.

**Spectral Foreground Fixer:** Figure 8 shows the qualitative analysis about the proposed SFF. The introduction of ControlNet [58] to provide subject information fails to alleviate the issue of foreground distortion, as distortions often occur on small areas and such design struggles to avoid the problem of encoding information loss. Our SFF achieves robust foreground preservation

| Method | PSNR↑ | SSIM↑ | CLIP-IS↑ | AS↑ |
|---|---|---|---|---|
| W/o fixer | 19.31 | 0.652 | 0.846 | 5.31 |
| ControlNet | 19.69 | 0.674 | 0.868 | 5.46 |
| SFF (Ours) | **25.53** | **0.831** | **0.946** | **6.84** |

Table 4: Different foreground fix strategies.

effects, as also substantiated by the quantitative results in Table 4. As mentioned above, SFF primarily works on critical small regions. While the refinement of these regions is important for visual quality, it has limited impact on global metrics. Thus, to better test the refinement effect, in Table 4 we crop small face regions for metric calculation.

**Handle Both Conditions:** We observe that thanks to the unified learning of text and image conditions in a single model, our DreamLight can generate results with the guidance of both conditions. As shown in Figure 9, our method can maintain the structure and elements of the given background while making additional adjustments based on text prompts. Note that this ability is emergent and our model does not undergo training for such situation.

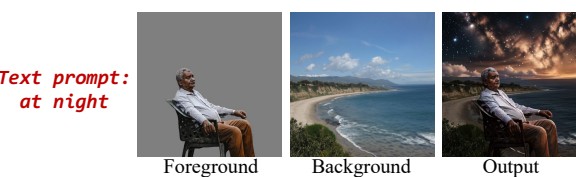

Text prompt: at night

| Foreground | Background | Output |

Figure 9: Visualization results of our DreamLight conditioned on both text prompt and background image.

## 5  Conclusion

In this paper, we present DreamLight that can perform both image-based and text-based relighting in a single model. In addition to extending application scenarios, our model emerges with the ability to handle both conditions simultaneously. By performing tendentious interaction between foreground and background in Position-Guided Light Adapter (PGLA), our model can achieve a natural and harmonious relighting effect. In addition, the Spectral Foreground Fixer (SFF) greatly enhance the consistency of the subject by adaptively leveraging information of different frequency bands. To train the model, we also develop a high-quality data generation pipeline. Experiment results have demonstrated that our DreamLight achieves excellent relighting performance and superior generalization ability for natural images. We hope this work could inspire more related researches and potential practical applications.

## Acknowledgements

This work was supported by Guangdong Natural Science Funds for Distinguished Young Scholar (No. 2025B1515020012).

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
