# OpenReview forum: "DreamLight: Towards Harmonious and Consistent Image Relighting"
_NeurIPS.cc/2025/Conference — NeurIPS 2025 poster_

### Official Review · Reviewer_6h7r · 2025-06-18

**Clarity:** 3
**Significance:** 3
**Originality:** 2
**Rating:** 3
**Confidence:** 3

**Summary:**

This paper proposes an image relighting method called DreamLight, which supports both image-based and text-based relighting. To generate more natural lighting effects between the foreground and background, the authors introduce a position-guided lighting adapter. Additionally, to avoid detail loss during VAE encoding, a spectral foreground fixer is proposed to preserve high-frequency details of foreground textures. However, some technical details require further explanation and experimental evaluation.

**Questions:**

My main concerns are summarized in Weaknesses Section. Considering some issues with the spectral foreground fixer and the confusion brought by ablation experiments, my initial rating to this paper is borderline reject. While I will consider my final rating according the feedback of the authors during the rebuttal section.

**Ethical Concerns:**

["NO or VERY MINOR ethics concerns only"]

**Limitations:**

The limitations are kindly discussed in the paper.

**Paper Formatting Concerns:**

None.

**Quality:**

2

**Strengths And Weaknesses:**

Strengths:
1. DreamLight supports both image-based relighting and text-based relighting.
2. A direction-biased mask attention mechanism is proposed to extract illumination from the four background directions (up, down, left, right), thereby achieving more accurate foreground relighting.
3. Qualitative and quantitative experimental results demonstrate that the proposed method achieves more natural and continuous lighting effects.

Weaknesses:
1. I noticed that the image-based relighting exhibits some incorrect shadow effects. Is it because the spectral foreground corrector directly replaces the foreground region with I_out, weakening the shadow interaction between the foreground and background?
2. The training data for the spectral foreground fixer is obtained using random color transformations, which significantly differ from actual lighting variations. Although ablation experiments demonstrate the module's effectiveness, what is the actual distribution of the predicted alpha and beta? Visual ablation experiments are needed to prove its specific role.
3. Why does the spectral foreground fixer predict beta parameters and directly add them to the high-frequency components? Its actual function and physical meaning are unclear.
4. After removing the spectral foreground fixer module, metrics such as PSNR were lower than IC-Light and IH. This is strange because the position-guided light adapter should better extract illumination information from four directions, and the performance metrics should be higher than these baselines. The authors need to further explain the reason.
5. The inference latency should be experimentally tested.

---

> ### Author Rebuttal · Authors · 2025-07-29
>
> We sincerely thank your insightful comments and valuable feedback. To the concerns expressed in Weaknesses and Questions, our response is as follows:
>
> ### **W1: The potential effect of SFF.**
> Thanks for your kind question.
> Since high-frequency components are primarily related to the detailed textures of the subject, the modifications made by SFF to high-frequency components are expected not to affect effects such as shadows.
> This is based on the assumption that the high-frequency components of an image correspond to the pixels varying drastically, such as object boundaries and textures, while the low-frequency components correspond to the high-level semantic information such as the color and light.
> Could you kindly describe what the "incorrect shadow effects" are? We are a bit confused about where the issue lies.
> Typically, our model can produce desirable light interaction effects in diverse scenarios.
> This is also confirmed by non-cherry-picked user study and extensive evaluations experiments.
> Certainly, there exists failure relighting cases in some complex scenarios for our model, but it may be unrelated to SFF. It is mainly because the initial relighting results generated by the diffusion model may have errors.
> We believe constructing more high-quality data in complex scenarios can further enhance the performance of our model.
>
> ### **W2: Discussion about SFF.**
> SFF is designed to preserve details of the foreground subject (such as the identity of small human faces). Actually, it can also be trained using real relighting data. Theoretically, if real relighting data has very strong subject consistency, SFF trained on it would achieve good performance. However, we do not use actual relighting data because they may contain samples where foreground details have changed. Specifically, a large amount of relighting training data is generated by models. Although we have utilized VLMs to filter low-quality data, there are still cases where foreground details have altered. Using such data for training may have adverse effects on SFF. Therefore, we directly use a self-supervised manner (color transformations) to construct pseudo relighting pairs with strongly consistent details for training SFF and find it works well.
>
> Following your kind suggestion, we print the numerical distributions of $\alpha$ and $\beta$ and compare them with the high-frequency part.
> Due to the restrictions of the official rules for this rebuttal, we are unfortunately unable to provide the distribution visualization images of these parameters for demonstration.
> Therefore, we present the statistical data of their distributions in the table below.
> $HQ _ {in}$ and $HQ' _ {in}$ denote the input and the adjusted
> high-frequency component, respectively. $HQ' _ {in} = HQ _ {in}*\alpha + \beta$. Mean, Min, and Max indicate the distribution of each parameter.  It can be seen that $\alpha$ and $\beta$ work together to adjust the distribution of high-frequency components, so as to adaptively enhance the details of the subject without affecting the relighting effects.
> Visual ablations of Figure 8 in the paper also illustrates the effectiveness of our SFF.
>
> |  | Mean | Min   | Max |
> |--------|--------|--------|--------|
> | $\alpha$ |  0.87  | 0.38  | 1.20 |
> |$\beta$ | 0.001 | -0.16  | 0.21   |
> |$HQ _ {in}$ | -0.001 | -0.81  | 1.04   |
> |$HQ'_ {in}$ | 0.001 | -0.75  | 1.14   |
>
>
> ### **W3: The meaning of $\beta$.**
> The design philosophy behind our SFF being engineered to output both $\alpha$ and $\beta$ is inspired by neural network layers. Since neural network layers often contain weights and biases, we intuitively adopted two predictive parameters and defined: $HQ' _ {in} = HQ _ {in}*\alpha + \beta$. Ideally, $\alpha$ is utilized to control the influence degree of initial high-frequency components and $\beta$ contains the adjustment information.
> Besides, the modulator is somewhat like a style transferrer, and the commonly used AdaLN operation also has scaling coefficients (our $\alpha$) and offset coefficients (our $beta$).
> We assume that these commonly used operations contain two parameters is because such approach can facilitate model learning.
>
> ### **W4: Performance metrics confusion.**
> There appears to be some misunderstanding here. The values you compared are likely from Table 4 (the ablation study on SFF) and Table 1 (the main results) in the paper. However, their test settings and metric calculation are actually different. As described in the paper from L304 to L307: "SFF primarily works on critical small regions. While the refinement of these regions is important for visual quality, it has limited impact on global metrics. Thus, to better test the refinement effect, in Table 4 we crop small face regions for metric calculation."
>
> Actually, when calculating the metrics for the entire image, the performance of the model without SFF (using only PGLA) is numerically close to that of the final version and significantly outperforms previous methods. The corresponding results are shown in the table below.
>
> | Method  | PSNR | SSIM   | LPIPS | CLIP-IS   |
> |--------|--------|--------|--------|-------|
> | INR |  16.79  | 0.694  | 0.231 | 0.872 |
> | PCT | 18.45 | 0.714  | 0.206   | 0.895 |
> | IH |  20.66  | 0.762  | 0.177  | 0.896 |
> | ICLight | 20.27  | 0.771 | 0.181  | 0.889 |
> | Ours w/o SFF | 22.09    | 0.781   | 0.162    | 0.907 |
> | Ours | 22.15  | 0.783  | 0.158  | 0.908 |
>
> ### **W5: Inference latency.**
> Thanks for your valuable advice!
> We provide the inference cost (FPS and memory utilization) of the proposed modules in the table below.
> All the value are obtained on a single H20 GPU.
> For comparison, we have also listed the relevant cost of ICLight.
> |  | Baseline | + PGLA   | + SFF (final model) | IC-Light   |
> |--------|:--------:|:--------:|:--------:|:--------:|
> | FPS |  0.585    | 0.491   | 0.411   | 0.303|
> | Memory (G) |  4.33    | 5.85   | 9.37   | 6.43|
>
> Moreover, to further demonstrate the inference latency of different parts of our model, we have counted the inference time of each module, and the results are as follows.
>
> |  | Matting | VAE  | PGLA | Diffusion   | Fixer (SFF)   |
> |--------|:--------:|:--------:|:--------:|:--------:|:--------:|
> | Time (s) |  0.32   | 0.29   | 0.92   | 0.31| 0.39|
>
> We will update these results in the manuscripts.
>
> Thanks again for your valuable time and insightful comments. We hope that our response can address your questions, and if you still have any concerns, we would be pleased to discuss them further with you.

---

> > ### Comment · Area_Chair_rWHU · 2025-08-07
> >
> > Dear 6h7r,
> >
> > We'd love to hear your thoughts on the rebuttal.
> > If the authors have resolved your (rebuttal) questions, please do tell them so. If the authors have not resolved your (rebuttal) questions, please do tell them so too.
> >
> > Thanks,
> > Your AC

---

> ### Author Response · Authors · 2025-08-06
>
> Dear Reviewer 6h7r,
>
> We sincerely thank you for your efforts in reviewing our paper and your valuable comments.
> May I kindly ask if we have fully addressed your concerns? If there are any remaining concerns, please feel free to raise them and we will try our best to explain.
>
> Sincerely,
>
> Authors

---

### Official Review · Reviewer_Vqyj · 2025-06-24

**Clarity:** 3
**Significance:** 3
**Originality:** 3
**Rating:** 4
**Confidence:** 4

**Summary:**

This paper introduces DreamLight, a universal image relighting framework that harmonizes foreground subjects with new backgrounds specified by images or text. It leverages pretrained diffusion models and proposes the Position-Guided Light Adapter (PGLA) to extract directional lighting from the background via masked attention, improving natural light interaction. A Spectral Foreground Fixer (SFF) further enhances appearance consistency.

**Questions:**

- Could you please clarify how your model performs in scenes with more than four dominant light directions? Are there any plans to extend the PGLA mechanism to support more flexible or adaptive directional queries?

- How sensitive is the Spectral Foreground Fixer to the quality and realism of its synthetic training data? Have you evaluated its effectiveness on real-world relighting scenarios?

- Would you be able to provide more details on the runtime performance, memory requirements, and inference speed of your method compared to existing baselines?

- Have you encountered any limitations or failure cases when the background contains strong lighting contrasts or complex shadows? If so, how does your method address these challenges?

**Ethical Concerns:**

["NO or VERY MINOR ethics concerns only"]

**Final Justification:**

Thanks for the response. The rebuttal addressed my concerns！

**Limitations:**

See Questions

**Quality:**

3

**Strengths And Weaknesses:**

Strengths:
- Writing is clear and figures support the technical explanations. The Position-Guided Light Adapter (PGLA) introduces a novel direction-biased masked attention mechanism that explicitly models multi-directional lighting cues, improving relighting quality. The Spectral Foreground Fixer (SFF) effectively addresses foreground distortion by adaptively combining high-frequency foreground details with low-frequency lighting from the background, enhancing visual fidelity.


Weaknesses:
- The directional light queries are limited to four fixed directions, which may restrict the model’s ability to handle complex lighting environments with multiple arbitrary light sources.
- The SFF training relies on synthetic color transformations that may not fully capture real-world lighting variations; additional evaluation on real lighting conditions would strengthen the claims.
- Computational cost and inference speed are not thoroughly discussed, which matters for practical deployment.
- Some ablations and deeper analysis on the relative importance of PGLA components and SFF would improve clarity.

---

> ### Author Rebuttal · Authors · 2025-07-29
>
> We sincerely thank your insightful comments and the appreciation of our work. To the concerns expressed in Weaknesses and Questions, our response is as follows:
>
> ### **W1&Q1: Light queries**
> Thanks for your valuable question!
> We set the light directions as "up, down, left, right" because the lighting information is derived from a 2D background image. In our view, the light sources in a 2D image tend to align with horizontal and vertical axes. Therefore, **these four directions serve as a natural "basis" for representing lighting orientations in 2D space**. Since light queries are high-dimensional features and there are multiple queries responsible for each direction (by default, 4 per direction, totaling 16), they are capable of representing diverse light information.
> Besides, the biased perception of each direction changes continuously (for example, the decay map responsible for leftward light gradually changes from 1 at the far left of the image to 0 at the far right).
> Direction-biased masked attention is a soft constraint rather than a hard distinction.
> High-dimensional light queries still have the ability to perceive multiple light sources and the foreground also interacts with all basic direction light features under the guidance of position. As a result, **more complex lighting patterns can be captured by organizing signals from these four base directions**.
> To prove that, we construct images featuring colors in 6 different directions as conditional background images for testing. Due to the restrictions of the official rules for this rebuttal, we are unfortunately unable to provide the generated results for illustration. We observe that the output relighted results can perceive such complex light source.
>
>
> We hope this explanation offers greater clarity. If you have any remaining questions, we would be glad to elaborate further.
> We will also refine the corresponding representations in the manuscript to make it clearer.
> As for extending the PGLA mechanism to support more flexible or adaptive directional queries, we consider this an interesting direction for exploration. It may contribute to more significant and natural light interaction effects in complex scenarios. However, how to enable them to obtain ideal information is quite challenging, which will be one of the directions for our future work. Thanks!
>
> ### **W2&Q2: Training set and evaluation of SFF**
> Thanks for your great question!
> SFF is designed to preserve details of the foreground subject (such as the identity of small human faces).
> Actually, it can also be trained using real relighting data. Theoretically, if real relighting data has very strong subject consistency, SFF trained on it would achieve good performance.
> However, we do not use actual relighting data because they may contain samples where foreground details have changed. Specifically, a large amount of relighting training data is generated by models. Although we have utilized VLMs to filter low-quality data, there are still cases where foreground details have altered. Using such data for training may have adverse effects on SFF. Therefore, we directly use a self-supervised manner (color transformations) to construct pseudo pairs with strongly consistent details for training SFF and find it works well.
>
> All our evaluation of SFF are conducted on real relighting scenarios. On the one hand, the quantitative evaluations in the paper are all performed on real relighted images. The demonstration figures in the paper (both image-based and text-based ones) are also results processed by SFF, which demonstrates that SFF can capture real light. On the other hand, we have also tested the relighting effects of images from real-world daily scenes, and SFF can maintain the details of the subject without compromising reasonable relighting effects.
>
> ### **W3&Q3: Computational cost and inference speed.**
> Thanks for your valuable advice!
> We provide the inference cost (FPS and memory utilization) of the proposed modules in the table below.
> All the value are obtained on a single H20 GPU.
> For comparison, we have also listed the relevant cost of ICLight.
> |  | Baseline | + PGLA   | + SFF (final model) | IC-Light   |
> |--------|:--------:|:--------:|:--------:|:--------:|
> | FPS |  0.585    | 0.491   | 0.411   | 0.303|
> | Memory (G) |  4.33    | 5.85   | 9.37   | 6.43|
>
> Moreover, to further demonstrate the inference latency of different parts of our model, we have counted the inference time of each module, and the results are as follows.
>
> |  | Matting | VAE  | PGLA | Diffusion   | Fixer (SFF)   |
> |--------|:--------:|:--------:|:--------:|:--------:|:--------:|
> | Time (s) |  0.32   | 0.29   | 0.92   | 0.31| 0.39|
>
> We will update these results in the manuscripts.
>
> ### **W4: Relative importance of PGLA and SFF.**
> PGLA and SFF have entirely different design purposes and functions. Specifically, PGLA is intended to enable the model to generate more reasonable light interaction effects in image-based relighting scenarios (serving to enhance relighting effects). In contrast, the role of SFF is to maintain the consistency of the subject, especially in regions with high information density such as small human faces (serving for ID preservation).
> They function in different aspects, so there is no comparison of which is more important. For instance, if only PGLA is used without SFF, the output image would have good relighting effects but the human face may be distorted. Conversely, if only SFF is utilized without PGLA, the model may produce results where foreground details are well preserved but the light effects are unreasonable.
>
> ### **Q4: Discussion about failure cases.**
> Typically, thanks to the PGLA module, which enables reasonable light interactions between the foreground and background, our model achieves great relighting effects even in scenes with strong lighting and shadows. However, there are indeed some failure cases in extremely complex scenarios. For example, when sunlight shines through multiple gaps in tree canopies, our model may struggle to generate all fully plausible light spots.
> We consider these scenarios to be highly challenging, and incorporating 3D-related representations might help address such cases. This will be one of the directions for our future improvements.
>
> Thanks again for your valuable time and insightful comments. We hope that our response can address your questions, and if you still have any concerns, we would be pleased to discuss them further with you.

---

> > ### Comment · Area_Chair_rWHU · 2025-08-07
> >
> > Dear Vqyj,
> >
> > We'd love to hear your thoughts on the rebuttal.
> > If the authors have resolved your (rebuttal) questions, please do tell them so. If the authors have not resolved your (rebuttal) questions, please do tell them so too.
> >
> > Thanks,
> > Your AC

---

> ### Author Response · Authors · 2025-08-06
>
> Dear Reviewer Vqyj,
>
> We sincerely thank you for your efforts in reviewing our paper and your insightful suggestions.
> May I kindly ask if we have fully addressed your concerns? If there are any remaining concerns, please feel free to raise them and we will try our best to explain.
>
> Sincerely,
>
> Authors

---

### Official Review · Reviewer_R5Ls · 2025-06-27

**Clarity:** 3
**Significance:** 3
**Originality:** 3
**Rating:** 5
**Confidence:** 4

**Summary:**

This paper proposes DreamLight, a method for relighting images based on either a text prompt or a background image. It introduces two main components: (1) the **Position-Guided Light Adapter (PGLA)**, a modified IP-Adapter designed to model lighting from four primary directions (left, right, up, down), and (2) the **Spectral Foreground Fixer (SFF)**, which enhances detail preservation by injecting high-frequency components from the original input into the relit result, improving areas like faces or text.

**Questions:**

## 1. Clarify Modulator Implementation in SFF

In Line 245, the paper states that SFF is fine-tuned on the VAE of Stable Diffusion 1.5. However, it’s unclear how the Modulator is implemented, given that it is supposed to take two inputs high-frequency foreground ($HQ_{in}$) and low-frequency relighting result ($LQ_{out}$) and output two modulation parameters ($\alpha$, $\beta$), whereas the VAE has only one single input and output which a single image.

**Question:** Could you clarify how the Modulator is structured and trained in this context?

## 2. Text Prompt Used in Image-Based Relighting

Since the model is based on Stable Diffusion 1.5, a text prompt is required at inference. However, the paper does not clearly state what prompt is used during image-based relighting. Figure 2 shows “Blend these two images,” but this is not explicitly mentioned in the text.

**Question:** What text prompt is used during inference for image-based relighting, and is it the same as during training? If not, how is the prompt selected, and how sensitive is performance to prompt choice?

## 3. Lack of Reflections or Subject-Light Interactions in Background
Supplementary Figure 1 shows a dog above a reflective surface (wet floor or water), but no reflection is present.

**Question:** Does the current pipeline, particularly SFF, suppress interactions such as reflections or shadows in the background? Would disabling SFF allow more light or reflection effects to propagate naturally?

**Ethical Concerns:**

["NO or VERY MINOR ethics concerns only"]

**Final Justification:**

The proposed technique demonstrates clear improvements in preserving fine details during relighting compared to prior methods, which making it worthy of acceptance.

The authors have adequately addressed my concerns regarding evaluation, ablation studies, and implementation details in their rebuttal. I have no remaining concerns.

**Limitations:**

The authors addressed some limitations in the supplementary material. However, one additional limitation worth mentioning is that the model does not handle physical interactions between the foreground and background, such as reflections or shadows, adapting to background surfaces. This could limit realism in certain scenarios.

**Paper Formatting Concerns:**

No concerns. The paper follows the NeurIPS template, stays within the 9-page content limit, and appears properly formatted.

**Quality:**

3

**Strengths And Weaknesses:**

## Strength:
- **Quality:**  The method is technically sound and well-executed, with clear contributions
- **Originality:** Both the Position-Guided Light Adapter (PGLA) and Spectral Foreground Fixer (SFF) present novel ideas tailored to the relighting task.
- **Significance:** The model effectively preserves foreground identity (e.g., face details), a common weakness in prior work. This makes it a strong candidate for real-world relighting applications and downstream tasks.
- **Clarity (high-level)**: The overall pipeline is well-structured and easy to follow.
- **Qualitative result:** The visual comparisons clearly demonstrate improved foreground consistency and lighting harmony over prior methods.
## Weakness:
- **Clarity (technical detail):**  Some implementation aspects, such as the implementation of the Modulator in SFF, are underexplained, limiting reproducibility.
- **Quantitative Evaluation:** The benchmark used for evaluation is internally created. While IC-Light did not release its test set, this raises concerns about potential cherry-picking or a lack of generalization.
- **Ablation gap:**  The ablation study on PGLA evaluates the impact of the spectral filter but omits a crucial variant: applying the filter without the light queries (i.e., IP-Adapter with filtering). This makes it difficult to disentangle the contributions of each component.

---

> ### Author Rebuttal · Authors · 2025-07-28
>
> We sincerely thank your insightful comments and the appreciation of our work. To the concerns expressed in Weaknesses and Questions, our response is as follows:
>
> ### **W1&Q1: Implementation details of SFF.**
> Thanks for your valuable advice!  Specifically, we combine the high-frequency foreground $HQ _ {in}$ and low-frequency relighting result $LQ _ {out}$ to form a single image as the input. To generate two modulation parameters, we modify the output channels of the last layer of the VAE to 6, and during initialization, the new channels copy the weights of original channels.
> We will revise the description of this part in the manuscript to make it clearer.
>
> ### **W2: Evaluation dataset.**
> Thanks for your valuable question!
> We understand your potential concerns.
> However,  there are indeed no widely used public benchmarks for relighting based on purely natural images.
> Existing public test sets are mostly based on environment maps.
> To enable testing on such dataset, we render the corresponding 2D background images of each view from the HDR panorama maps based on the relevant information (FOV, camera extrinsic parameters) provided by the dataset.
> Take TensoIR as an example, we calculate the metrics on all test views with the scenarios of provided HDR maps  and the results are shown in the table below.
> In fact, the locally rendered background images (just patches in the scenarios) in this way are actually unable to provide sufficient light information (there is a huge gap compared with the information that HDR panorama maps can offer). Therefore, there is a discrepancy between the prediction results of the evaluated models and the ground truth provided by the dataset. However, we observe that under such background conditions, previous methods also expose some problems, such as color bleeding of ICLight and weak light changes in INR. Generally, our model performs better in most cases.
>
>
> | Method  | PSNR | SSIM   | LPIPS |
> |--------|--------|--------|--------|
> | INR | 17.61   | 0.683 | 0.327  |
> | PCT | 18.59   | 0.705  | 0.296  |
> | IH |  19.13  | 0.726  | 0.214  |
> | ICLight | 18.45    | 0.684   | 0.236 |
> | Ours | 20.86    | 0.783   | 0.175|
>
>
> Besides, our test set is indeed not deliberately cherry-picked. When creating the test set, manual intervention only aimed to ensure diverse scenes (for evaluating generalization ability) and natural ground-truth results, without referring to the outcomes of our model. Additionally, we will make our test set publicly available to benefit the community.
>
> ### **W3: Ablation gap.**
> Thanks for your helpful suggestion!
> We have experimented with the ablation of IP-Adapter with filtering.  Due to the restrictions of the official rules for this rebuttal, we are unfortunately unable to provide visual images for demonstration. Therefore, we present the results of quantitative comparisons below and describe the visualization comparison through text. Specifically, as shown in the results in the table below, applying the filter without the light queries (i.e., IP-Adapter with filtering) does not bring significant performance improvement. Visualization results also show that the relighting effects of IP-Adapter with filtering are similar to those of the vanilla IP-Adapter, with no obvious light interaction effects.
> This is because although filtering is introduced to enhance the representation of high-level light information in the background, IP-Adapter with filtering, like the vanilla IP-Adapter, performs unguided interactions between the subject and the background, resulting in light tending to have a general effect. In contrast, through the utilization of direction-biased masked attention and light queries, our PGLA selectively transmits background lighting information, enabling the foreground to acquire lighting that is harmonious with the background.
>
> | Method  | PSNR | SSIM   | LPIPS | CLIP-IS   |
> |--------|--------|--------|--------|-------|
> | w/o adapter |  18.58   | 0.732   | 0.221   | 0.865 |
> | vanilla IP-Adapter | 20.23  | 0.752   | 0.184    | 0.891 |
> | IP-Adapter with filtering |  20.31 | 0.759 | 0.181    | 0.893 |
> | w/o Filter | 21.81   | 0.778   | 0.162   | 0.903 |
> | PGLA (Ours) | 22.15  | 0.783 | 0.158 | 0.908 |
>
> ### **Q2: Text prompt.**
> Thanks for your valuable question! As mentioned in L118 in the main paper, the text prompt is set to "blend these two images" for image-based relighting. This is the same for both training and inference by default. We will make the corresponding descriptions clearer.
> In addition, our model also has the generalization ability for other text prompts, such as "natural light". However, since "blend these two images" is used during training, this prompt yields the best results in most inference cases.
>
> ### **Q3: Reflections or subject-light interactions in background.**
> Great question! In fact, this is somewhat related to the definition of the task. For image-based relighting, we follow the definition of this setting in previous works: only modifying the foreground region to adapt to the background while keeping the background region unchanged.
> The training objectives and corresponding training data are also organized in this setting.
> Hence, in examples like the one in Figure 1 of the supplementary materials you mentioned, there is no reflection of the dog in the background. In contrast, for text-based relighting, since the background is also generated by the model, there are interaction effects between the foreground and background (such as shadows), as demonstrated in the examples in Figure 3 of the supplementary materials.
> Our pipeline and the proposed SFF do not suppress interactions such as reflections or shadows in the background region.
>
> Besides, we believe this question highly insightful. If the background region in image-based relighting could also exhibit reasonable interaction effects, the overall result would be more natural and realistic. However, this also poses additional challenges, such as how to obtain such training data in large quantities. This will be a direction for our further optimization in the future.
>
> Thanks again for your valuable time and insightful comments. We hope that our response can address your questions, and if you still have any concerns, we would be pleased to discuss them further with you.

---

> > ### Comment · Reviewer_R5Ls · 2025-08-03
> > **Appreciated Response – No Remaining Concerns**
> >
> > Thank you for the detailed response and clarification.
> >
> > This work presents a technically novel relighting method that effectively preserves high-quality details, which I believe makes it worthy of acceptance.
> >
> > While the evaluation would be better if conducted on a public benchmark rather than an author-proposed dataset, I understand the lack of available public test sets. Therefore, this alone should not be a reason for rejection.
> >
> > **Current rating:** 5 – Accept

---

> > > ### Author Response · Authors · 2025-08-04
> > >
> > > Thanks for your response. We are glad that our rebuttal is able to address your concerns. We sincerely appreciate your review time and the valuable comments. We will attach great importance to revising the paper according to your suggestions.

---

### Official Review · Reviewer_X4Rx · 2025-07-02

**Clarity:** 3
**Significance:** 3
**Originality:** 3
**Rating:** 5
**Confidence:** 4

**Summary:**

This paper proposes a new method called DreamLight to perform both image-based and text-based relighting, given a foreground image and a background image. Two key components are introduced: the position-guided light adapter and the spectral foreground fixer. The authors also generate their own relighting data for the training of the model. Experiments are conducted on the generated data, and DreamLight is compared against methods like IC-Light.

**Questions:**

1. The proposed method is evaluated on an author-generated dataset. What's the performance of the method on existing datasets that are used as benchmarks for relighting, such as TensoIR (https://github.com/Haian-Jin/TensoIR)?

2. Related to Q1, most results shown in this paper are portraits. How does the method perform on relighting general objects with different materials?

3. The design of the direction-biased masked attention is pretty interesting. On the other hand, I don't have a good understanding of why it is working well. Why is it important to have four fixed directions? What happens if you need more complex lighting patterns? Or maybe I'm missing something here.

4. While Table 4 shows the performance trade-off of adding the fixer to the relighting pipeline, what about the time cost?

5. Line 214: The modulator M is trained with pseudo pairs of relighting data. Why not use actual relighting data to train it?


Overall, the proposed method has interesting designs to solve a challenging task, while there are some missing pieces in fully understanding the effectiveness of the method. If the authors can address the questions mentioned above, I can raise my score.

**Ethical Concerns:**

["NO or VERY MINOR ethics concerns only"]

**Final Justification:**

After reviewing the rebuttal, I'm satisfied with the authors responses. Overall, this is an interesting approach for relighting with some novel designs that can inspire other work. I recommend the acceptance of this paper.

**Limitations:**

No discussion on the limitations.

**Quality:**

3

**Strengths And Weaknesses:**

### Strengths
1. The idea of having a unified approach for both image-based and text-based relighting makes sense. The proposed method, DreamLight, seems to be a reasonable solution to this task.
2. The proposed Position-Guided Light Adapter is a novel design that is tailored to model lighting information in an effective format. See below for additional questions on this part for clarification.

### Weaknesses
1. One major weakness is the lack of experiments on the existing relighting datasets. Currently, the method is trained and tested on the author-generated data. It will be more convincing if the method is also tested on public benchmarks.
2. While the Position-Guided Light Adapter is directly designed for the relighting task, the second proposed component, Spectral Foreground Fixer, seems to be less tailored for relighting. For example, some lighting effects are actually high-frequency such as specular reflection. It is unclear if the proposed fixer can handle such a relighting situation.
3. Some clarification is needed to fully understand the effectiveness of the DreamLight method (see questions below).

### Typo
- Table 2, 'PP denotes PowerPoint'--it should be PowerPaint.

---

> ### Author Rebuttal · Authors · 2025-07-28
>
> We sincerely thank your insightful comments and the appreciation of our work. To the concerns expressed in Weaknesses and Questions, our response is as follows:
>
> ### **W1&Q1: Evaluation on public datasets.**
> Thanks for your valuable advice!
> Our model and the compared works all rely on pure natural images rather than HDR maps for providing light information.
> To the best of our knowledge, there are indeed no widely used public benchmarks for relighting based on **purely natural background images**.
> The benchmark  you recommended (TensorIR) also **relies on HDR panoramas** to provide light information rather than mere 2D natural background images.
>
> To enable testing on this dataset, we render the corresponding 2D background images of each view from the HDR maps based on the relevant information (FOV, camera extrinsic parameters) provided by the dataset.
> We calculate the metrics on all test views with the scenarios in "high_res_envmaps_1k" folder  and the results are shown in the table below.
> In fact, we find that the locally rendered background images (just patches in the scenarios) in this way are actually unable to provide sufficient light information (there is a huge gap compared with the information that HDR panorama maps can offer). Therefore, there is a discrepancy between the prediction results of the evaluated models and the ground truth provided by the dataset, leading to generally low overall scores. However, we observe that under such background conditions, previous methods also expose some problems, such as color bleeding of ICLight and weak light changes in INR. Generally, our model performs better in most cases.
>
>
> | Method  | PSNR | SSIM   | LPIPS |
> |--------|--------|--------|--------|
> | INR | 17.61   | 0.683 | 0.327  |
> | PCT | 18.59   | 0.705  | 0.296  |
> | IH |  19.13  | 0.726  | 0.214  |
> | ICLight | 18.45    | 0.684   | 0.236 |
> | Ours | 20.86    | 0.783   | 0.175|
>
>
> Furthermore, we understand your potential concerns, yet our test set is indeed not deliberately cherry-picked. When creating the test set, manual intervention only aimed to ensure diverse scenes (for evaluating generalization ability) and natural ground-truth results, without referring to the outcomes of our model. Additionally, we will make our test set publicly available to benefit the community.
>
>
>
>
> ### **Q2: Relighting effect on general objects with different materials.**
> Thanks for your great question!
> Due to the restrictions of the official rules for this rebuttal, we are unfortunately unable to provide relevant images of relighting results for demonstration. The reason why the paper mainly presents portraits is that some of the comparison methods are mainly trained on portrait data. Actually, our model has great generalization ability for general objects with different materials.
>
> On the one hand, as shown in Figure 5 of the supplementary materials, our training data contains, and even includes a large amount of, samples of general objects. Specifically, the ratio of portrait data to non-portrait data is about 3:2, which ensures the effectiveness of our model in relighting general objects.
> On the other hand, the benchmark (TensoIR) mentioned in Q1 is related to general objects of different materials, and the relevant results demonstrate the superiority of our method over previous approaches in handling general objects. In addition, we have also constructed a new test set consisting of 100 images of pure general objects (without portraits) and compared the performance. The results below also prove the effectiveness of our method.
>
> | Method  | PSNR | SSIM   | LPIPS | CLIP-IS   |
> |--------|--------|--------|--------|-------|
> | INR |  20.87    | 0.722   | 0.227   | 0.881|
> | PCT | 21.48    | 0.733  | 0.204   | 0.893 |
> | IH |     23.29     |  0.797 | 0.175  | 0.896 |
> | ICLight | 23.61  |  0.805 |  0.156  | 0.902 |
> | Ours | 24.57  | 0.816  | 0.132   | 0.917|
>
> ### **W2: Spectral Foreground Fixer seems to be less tailored for relighting.**
> Thanks for your insightful question.
> SFF is  designed to  preserve details of the foreground subject (such as the identity of small human faces) while minimizing the impact of reorganizing frequency components on relighting effects.
> Thus its design also takes into account factors related to relighting effects.
> Specifically, directly combining the high-frequency components of the input foreground with the low-frequency components of the initial relighting result may cause artifacts and biased lighting in some scenarios.
> To generate more consistent and natural results, we propose to train a modulator to adaptively adjust different parts.
> This enables SFF to have good generalization ability for diverse light.
>
> High-frequency light effects (such as specular reflections) often appear in regions with low information density (e.g., non-facial regions). Since SFF can adaptively adjust the influence degree of the high-frequency components, it will make more use of the relighting information in such cases, thus achieving desirable results.
> We have tested such cases where smooth object surfaces are exposed to intense light.
> The results show that the relighting outcomes are satisfactory, and SFF does not have a negative impact on such scenes.
> However, in some cases (such as when there are high-frequency light spots on a small face), SFF may affect the relighting and soften the light effect. This is because  there exists a trade-off between them. Users can control this by adjusting the kernel size of the wavelet transform. How to better improve both aspects simultaneously for such challenging scenarios is also an important direction for our future work.
>
>
> ### **W3&Q3: Further explanation about PGLA.**
> The reason for setting "up, down, left, right" light directions is that the light information is provided by a 2D background image. We assume that the light sources in a 2D image are mainly in the horizontal and vertical directions. Therefore, the four directions (up, down, left, right) can form the "basis" for the light directions in a 2D image.
>
> Besides, the biased perception of each direction changes continuously (for example, the decay map responsible for leftward light gradually changes from 1 at the far left of the image to 0 at the far right).
> The foreground also interacts with all basic direction light features under the guidance of position.
> Thus, other types of light patterns can be automatically perceived through the combination of information from the up, down, left, and right directions.
> For instance, a light source in the upper-left corner corresponds to strong perceptions of leftward and upward light.
> We wonder if this explanation is clearer. If you have any further doubts, we are happy to provide further clarification.
>
> ### **W3&Q4: Time cost of SFF.**
> Thanks for your valuable advice!
> We provide the inference cost (FPS and memory utilization) of the proposed modules in the table below.
> All the value are obtained on a single H20 GPU.
> For comparison, we have also listed the relevant cost of ICLight.
> |  | Baseline | + PGLA   | + SFF (final model) | IC-Light   |
> |--------|:--------:|:--------:|:--------:|:--------:|
> | FPS |  0.585    | 0.491   | 0.411   | 0.303|
> | Memory (G) |  4.33    | 5.85   | 9.37   | 6.43|
>
> We will update these results in the manuscripts.
>
> ### **W3&Q5: Training data of SFF.**
> Thanks for your great question!
> We do not use actual relighting data because they contain samples where foreground details have changed. Specifically, a large amount of relighting training data is generated by models. Although it undergoes screening, there are still cases where foreground details have altered. Using such data has adverse effects on SFF, causing its learning objectives to deviate. Therefore, we directly use a self-supervised manner to construct pseudo pairs with strongly consistent details for training SFF.
>
> ### **Typo**
> Thanks! We have carefully checked the manuscript and corrected the typos.
>
> Thanks again for your valuable time and comments. We hope that our response can address your questions, and if you still have any concerns, we would be pleased to discuss them further with you.

---

> ### Author Response · Authors · 2025-08-06
>
> Dear Reviewer X4Rx,
>
> We sincerely thank you for your efforts in reviewing our paper and your valuable feedback.
> May I kindly ask if we have fully addressed your concerns? If there are any remaining concerns, please feel free to raise them and we will try our best to explain.
>
> Sincerely,
>
> Authors

---

> > ### Comment · Reviewer_X4Rx · 2025-08-06
> >
> > Thanks for the response. The rebuttal addressed my concerns. Please add the experiments and clarification to the final version of the paper. I recommend the acceptance of the paper.

---

> > > ### Author Response · Authors · 2025-08-07
> > >
> > > We sincerely appreciate your support and encouragement. We will incorporate the discussions mentioned above into our revised manuscript.

---

### Comment · Area_Chair_rWHU · 2025-08-04

Dear Reviewers,

Please take a look at the rebuttal and check if your questions and concerns have been addressed and everything is clear. Now is the time to clarify any remaining issues in discussion with the authors.

Thanks,
Your AC

---

### Decision · Program_Chairs · 2025-09-17

**Decision:**

Accept (poster)

**Comment:**

The paper proposes an approach for relighting objects or portraits given a text prompt or background image. To that end it introduces new modules for extracting directed lighting information from the background image and for refining the foreground of the harmonized image.

The reviewers appreciated the strong experimental results, clear writing, and novelty of the proposed components. They were concerned about missing experiments on existing datasets, on objects, and to further ablate design choices. For the refinement module, they noted missing implementation details, an intuitive explanation on its functioning, and questioned the use of synthetic data for training.
Results and technical details provided in the rebuttal and discussion addressed these concerns sufficiently and swayed 3 of the reviewers towards accepting the paper. The remaining reviewer with a borderline reject did not respond to the rebuttal and discussion, but the AC concludes that the rebuttal sufficiently addresses the raised questions. The AC agrees with the recommendation of the reviewers.